# Directionally-Grown Carboxymethyl Cellulose/Reduced Graphene Oxide Aerogel with Excellent Structure Stability and Adsorption Capacity

**DOI:** 10.3390/polym12102219

**Published:** 2020-09-27

**Authors:** Mengke Zhao, Sufeng Zhang, Guigan Fang, Chen Huang, Ting Wu

**Affiliations:** 1Shaanxi Provincial Key Laboratory of Papermaking Technology and Specialty Paper Development, National Demonstration Center for Experimental Light Chemistry Engineering Education, Key Laboratory of Paper Based Functional Materials of China National Light Industry, Shaanxi University of Science and Technology, Xian 710021, China; mengkezhao18@163.com; 2Institute of Chemical Industry of Forest Products, Chinese Academy of Forestry, Nanjing 210042, China; huangchen3127@njfu.edu.cn (C.H.); wuting@icifp.cn (T.W.)

**Keywords:** carboxymethyl cellulose, reduced graphene oxide, directional freezing technique, adsorption, wastewater treatment

## Abstract

A novel three-dimensional carboxymethyl cellulose (CMC)/reduced graphene oxide (rGO) composite aerogel crosslinked by poly (methyl vinyl ether-*co*-maleic acid)/poly (ethylene glycol) system via a directional freezing technique exhibits high structure stability while simultaneously maintaining its excellent adsorption capacity to remove organic dyes from liquid. A series of crosslinked aerogels with different amounts of GO were investigated for their adsorption capacity of methylene blue (MB), which were found to be superb adsorbents, and the maximum adsorption capacity reached 520.67 mg/g with the incorporation of rGO. The adsorption kinetics and isotherm studies revealed that the adsorption process followed the pseudo-second-order model and the Langmuir adsorption model, and the adsorption was a spontaneous process. Furthermore, the crosslinked aerogel can be easily recycled after washing with dilute HCl solution, which could retain over 97% of the adsorption capacity after recycling five times. These excellent properties endow the crosslinked CMC/rGO aerogel’s potential in wastewater treatment and environment protection.

## 1. Introduction

With the rapid development of the social economy, water pollution caused by wastewater from various industries is emerging as an intractable and vital issue for modern society. Organic contaminants, such as organic dyes and solvents, are the leading contributors to water pollution and are nonbiodegradable, unable to be metabolized in living systems and can cause destructive effects to the living organisms, such as allergic reaction problems, carcinogenic effects and heart malfunctions [1,2,3,4]. Therefore, such contaminants need to be removed with low-cost competent techniques in order to use the water for drinking or other purposes, which is growing to be a foremost challenge currently. Adsorption [5,6,7], an economic, efficient and eco-friendly technology, has the characteristics of clear principle and convenient industrial application.

Recently, aerogels have drawn significant research attention in the adsorption field due to their superior performance of low density, high specific surface area, high porosity and recyclability [8,9,10]. A lot of materials such as graphene [11], cellulose and its derivatives [12], metal organic frameworks [13], clays [14], zeolite [15], etc., have been used to fabricate aerogel adsorbents for wastewater purification, and they adsorb dyes through electrostatic interaction, π–π stacking, hydrogen bonding interaction and Van der Waals’ force [16]. Among them, graphene and its composite materials have been reported as a superior adsorbent for removal of dyes from wastewater because of the high surface area, chemical stability and large delocalized π-electron arrangements. However, graphene has some drawbacks, such as poor dispersion and agglomeration [9]. Graphene oxide (GO), one of the most important graphene derivatives, is a two-dimensional (2D) nanomaterial obtained by chemical oxidation and exfoliation of natural graphite. GO could disperse readily in aqueous media and has excellent surface activity in most polar solvents because of the abundant oxygen-containing groups, such as carboxyl, hydroxyl and epoxide groups. However the structure stability of pure GO aerogel is poor in aqueous conditions, which leads to great recycling difficulties and causes secondary pollution [17]. Until now, great efforts have been put forward to fabricate a composite aerogel containing GO and polymers with superior properties, which have been employed for efficient removal of toxic dyes and metal ions from wastewater [18]. For example, Chen et al. [19] synthesized a reusable 3D agar/GO composite aerogel by a vacuum freeze-drying method and evaluated its potential as an adsorbent of methylene blue (MB). Cheng et al. [20] prepared a bead-like magnetic GO/poly (vinyl alcohol) composite aerogel in one-step and found it exhibited adsorption capacity of 231.12 mg/g toward MB. Liao et al. [18] designed a polydopamine modified GO/chitosan aerogel, and the composite aerogel showed high adsorption capacity for uranium (VI), with ability up to 415.9 mg/g at a pH value of 6.0. Tang et al. [21] reported a self-recoverable, adjustable amphiphilic graphene aerogel as an efficient and recyclable absorbent for dye removal. All these articles showed great potential of GO as an additive to enhance the adsorption function of the generated materials.

Carboxymethyl cellulose (CMC), a biopolymer with a large number of carboxymethyl groups, has been considered as an excellent low-cost matrix material to impart certain mechanical strength [22,23]. In recent years, CMC-based aerogels have shown considerable potential use as absorbents, electrochemical supercapacitors, flexible sensing materials and drug delivery due to their solubility, degradability and biocompatibility. However, the abundant carboxymethyl groups make them very hydrophilic and thus limit their practical applications in aqueous environments. Covalent and non-covalent crosslinking methods are thus proposed to enhance the structure stability of the CMC-based materials [24,25]. Recently, the combination of poly (methyl vinyl ether-alt-maleic acid) (PMVEMA) and polyethylene glycol (PEG) with cellulosic materials has attracted attention. Aerogels crosslinked by a PMVEMA/PEG system have been used in dye removal and medical/pharmaceutical fields [26,27,28]. Moreover, a directional growth method wa proposed to produce a highly porous cellulose nanofibres/graphene oxide/sepiolite nanorod foam by Bernd Wicklein at al. [29], and the foam was mechanically stiff in the freezing direction and was able to sustain a considerable load. In a similar way, Liang at al. [30] formed various nanocellulose foams crosslinked with PMVEMA and PEG by both the directional and un-directional freezing techniques. The highly organized structures in the achieved foams exhibited enhanced mechanical performance and showed excellent water stability. After that, the authors used the foams for MB adsorption [31], and it showed excellent adsorption capacity, structure stability and reuse performance.

Inspired by the aforementioned research, herein, a novel CMC/rGO composite aerogel crosslinked by the PMVEMA/PEG system was fabricated in an eco-friendly way via the template-directed growth, and it exhibited strong structure stability in aqueous conditions and excellent adsorption capacity to remove organic dyes. The physical and chemical properties of the aerogel were analyzed with SEM, FT-IR, XRD, XPS and Raman spectra. We demonstrated that the synthesized CMC/rGO aerogels could remove MB from aqueous solution substantially. The effects of GO addition, initial concentration and recycling performance on the adsorption efficiency were systematically investigated with batch adsorption experiments. Moreover, we thoroughly studied the equilibrium, kinetics, adsorption mechanism and desorption process. These features revealed that the template-directed grown crosslinked CMC/rGO aerogel could be a high-efficiency adsorbent for the removal of MB from aqueous solution.

## 2. Materials and Methods

### 2.1. Chemicals

Carboxymethyl cellulose sodium (CMC) was purchased from Guilin Qihong Technology Co., Ltd. (Guilin, Guangxi, China). Graphene oxide (GO) was obtained from Nanjing XFNANO Materials Tech Co., Ltd. (Nanjing, China). Poly (methyl vinyl ether-alt-maleic acid) (PMVEMA) was purchased from Sigma-Aldrich Chemical Reagent Co., Ltd (St. Louis, MO, USA). Polyethylene glycol (PEG), with a *M*_w_ of 4000 g/mol, was purchased from Shanghai Aladdin Biochemical Technology Co., Ltd. (Shanghai, China). Methylene blue (MB) and hydrochloric acid (HCl) were acquired from Tianjin Chemical Reagent Research Institute Co., Ltd. (Tianjin, China). All the chemicals were used as received without any purification.

### 2.2. Preparation of Crosslinked CMC/rGO Aerogel

The crosslinked CMC/rGO aerogel was prepared according to the steps in Figure 1. First, CMC powder was dissolved in DI water (2 wt %) at room temperature (RT) with agitation for 12 h. The GO suspension (~2.5 g/L) was obtained by dispersing in DI water with agitation, which was then exfoliated in ultrasound for 30 min, these steps of agitation and ultrasound were repeated three times, and finally the GO suspension was obtained by centrifugation. After that, the two solutions were mixed at the weight ratio of 1:0.01, 1:0.03, 1:0.05, 1:0.07 and 1:0.1, and the mixture was then stirred magnetically for 1 h. Crosslinker was prepared by dissolving PMVEMA:PEG (6.00 g) in DI water (20.00 mL) at the mass ratio of 6.7:1 under magnetic stirring at 90 °C for 1 h (30 wt %). Then, a certain amount of the as-prepared crosslinker was added into the CMC/GO solution with magnetic stirring for 6 h to prepare a series of crosslinked CMC/GO composites. Finally, the mixture was transferred into glass vials and then placed on the top of a copper rod in contact with a liquid-nitrogen bath for about 6 min and then freeze-casted at −70 °C to obtain the crosslinked CMC/GO (L-CMC/GO) aerogels. Finally, samples were cured at 105 °C for 12 h to obtain crosslinked CMC/rGO (L-CMC/rGO) aerogels [32]. All the samples were stored in a desiccator prior to the test.

### 2.3. Characterization

Fourier-transformed infrared (FTIR) spectra of CMC, GO CMC/GO, L-CMC/GO and L-CMC/rGO aerogels were recorded with a FT-IR spectrometer (Bruker Vertex70, Karlsruhe, Germany) over the wavenumber range of 4000–500 cm^−1^. Scanning electron microscopy (SEM) images were obtained using a scanning electron microscope (ESCALAB 250Xi, Thermo Fisher Scientific, Waltham, MA, USA). X-ray diffraction (XRD) measurements were carried out on a Bruker D8 Venture instrument (Bruker, Karlsruhe, Germany) with Cu Kα radiation (λ = 0.15418 nm) as X-ray source at 40 kV accelerating voltage and 30 mA current. X-ray photoelectron spectroscopy (XPS) was collected on a Shimadzu AXIS UltraDLD instrument (Shimadzu, Kyoto, Japan) with Al Kα X-ray source to measure the chemical characteristics of the samples. The structure stability of the CMC/GO and crosslinked aerogels in liquid were assessed through immersing the samples into distilled water with shaking for 48 h. Compression fatigue tests of the aerogels were performed using a universal mechanical tester (Shimadzu AGS-X series, Kyoto, Japan) equipped with a 500 N load cell at RT. Raman spectra were conducted at RT using LabRAM HR Evolution instrument (Horiba Jobin Yvon, Paris, France). The spectra consist of two well-known characteristic GO bands at ~1350 cm^−1^ (D band) and ~1590 cm^−1^ (G band), respectively. UV-Vis spectroscopic measurements were carried out to determine the concentrations of MB in supernatant solutions by referring to the standard curve of MB at the maximum wavelength (664 nm) of MB dye.

### 2.4. Dye Adsorption Experiments

#### 2.4.1. Effects of GO Addition and Initial Dye Concentration on the Adsorption Efficiency

GO contains many hydroxyl functional groups resulting in a hydrophilic property. In order to study the effect of GO addition on the adsorption capacity, five L-CMC/rGO aerogels with different GO ratios were subjected to adsorption tests for MB dye under the optimized conditions (adsorbent dose, 0.03 g; temperature, 25 °C; adsorption time, 12 h). In a typical adsorption process, aerogel was added to 40 mL of 100 ppm MB solution in a 100 mL erlenmeyer flask, which was placed in a water bath. At the completion of preset time intervals, sample was collected and the residual amount of dye in the solution was detected by UV-Vis spectrometry at 664 nm for calculating the adsorption efficiency. The effects of initial concentration on the MB adsorption capacity of the aerogels were investigated by adjusting the concentration in the range of 100–500 ppm.

#### 2.4.2. Kinetic and Isothermal Adsorption Model

For kinetic and isothermal adsorption model study, the aerogels were immersed into 40 mL MB solution of 100, 200, 300, 400, 500 ppm at RT. At the completion of preset time intervals, the current concentration was measured using a UV-Vis spectrophotometer. The adsorption capacity of the aerogels at time t (*q*_t_, mg/g) and the equilibrium adsorption capacity (*q*_e_, mg/g) were calculated based on the changes of the MB concentration before and after adsorption, according to the following equation, respectively:(1)$$q_{t}=\frac{\left(C_{0}-C_{t}\right)\;V}{m}$$
(2)$$q_{e}=\frac{\left(C_{0}-C_{e}\right)\;V}{m}$$
where *C*_0_ and *C*_t_ (mg/L) are the concentrations of MB solution at the initial time and time *t* (h), respectively; *C*_e_ (mg/L) is the equilibrium concentration of MB solution; *V* (L) is the volume of the MB solution; and *m* (g) is the quality of aerogel.

### 2.5. Desorption and Reuse Experiments

The dye-loaded aerogels were collected and thoroughly washed in D.I. water. Next, they were immersed in 40 mL of desorbent solutions containing 0.1 mol/L HCl and the suspensions were shaken for 1 h. Subsequently, the aerogels were washed with 40 mL D.I. water. In order to ensure the adsorbents were eluted completely, the above steps were repeated three times. After that, the aerogels were recovered by freeze-drying for the next cycle.

## 3. Results and Discussion

### 3.1. Characterization

The SEM images of the prepared L-CMC/rGO aerogels are presented in Figure 2. The results indicated that all the samples showed the same porous microstructure with aligned tubular pores and a hole-shaped wrinkle structure. Moreover, SEM images demonstrated GOs, which increased the number of hydrophilic groups to enhance the liquid diffusion, and were efficiently dispersed within the matrix, considering no clusters or agglomerates were observed. The 3D multi-pore structure provided not only the high surface area but also the large mass transfer channels, which may contribute to the high adsorption capacity for dyes.

The chemical structure of the aerogels was analyzed with FTIR, XRD XPS and Raman. In the FTIR spectra (Figure 3a), pure CMC showed the characteristic peaks at 3335, 1591, 1417, 1324 and 1054 cm^−1^ that corresponded to the stretching vibrations of O–H, asymmetric and symmetric stretching vibrations of C=O, the bending vibration of C–H and the bending vibration of C–O–C [33], respectively. Pure GO showed the strong bands at 3438, 1721, 1625 and 1054 cm^−1^, which were related to the stretching vibration of O–H, C=O, C=C and C–O–C groups [19], respectively, while for the CMC/GO aerogel, the characteristic band of C=O of GO can be observed but is not obvious, which revealed the successful introduction of GO. That the peak was not obvious may be due to the low amount of GO. The L-CMC/rGO aerogel (reduced after curing at 105 °C for 12 h) crosslinked by a PMVEMA/PEG system [30] (Figure 4) was previously reported to occur through an esterification reaction [34]. The available carboxylic acid groups on the PMVEMA were crosslinked with the hydroxyl group on the CMC/GO surface or the terminal hydroxyl groups of PEG to establish the ester linkages.

The formation of ester cross-linkages was confirmed by comparing the FTIR spectra of a 0.1 M NaOH treated sample with an untreated sample, as summarized in Figure 3a.

The peaks observed at 1715 cm^−1^ of untreated sample can be due to both the ester linkage and the untreated carboxylic acid groups. In the NaOH treated sample, the peak at 1589 cm^−1^ was attributed to the carboxylate formed because of the neutralization of the unreacted carboxylic acid groups and the sodium hydroxide, and the remaining signal at 1715 cm^−1^ solely confirmed the establishment of ester cross-linkages between cross-linker and CMC/GO, which were previously reported by Goetz et al. [27,31].

The XRD pattern of the prepared aerogels and their initial components are displayed in Figure 3b. As for the raw materials, CMC exhibited the main peaks at 2θ = 14.79° and 22.44°; GO showed a strong and sharp diffraction peak at 2θ = 9.63° [33]. After the aerogel formation, the characteristic peaks of GO were found transferred to 2θ = 8.84° (patterns of CMC/GO and L-CMC/GO aerogels), which was mainly due to the intercalation of CMC into the layers of GO sheets. For L-CMC/rGO aerogels (reduced after curing at 105 °C for 12 h), the peak of GO disappeared due to the removal of the oxygen-containing groups and the partial reduction of the GO. In addition, the peak at 2θ = 22.44° of CMC transferred to 22.16° and became much stronger, which was attributed to the more regular structure after reduction cure at high temperature [35].

Figure 3c shows the Raman spectra of GO, CMC/GO and crosslinked aerogels before and after reduction. The D band is located at ~1354 cm^−1^ and the G band is at ~1596 cm^−1^ [36], which originated from the breathing mode of κ-point phonons of A_1g_ symmetry and the E_2g_ phonon of sp^2^ carbon atoms, respectively. Compared with GO, CMC/GO and L-CMC/GO aerogels showed higher I_D_/I_G_ (1.0052 and 1.0055, respectively), which can be attributed to the insertion of CMC and PMVEMA/PEG into the GO nanosheets. After reduction, it decreased from 1.0055 to 0.7048, which indicated an increase in the average size of the in-plane sp^2^ domains and a decrease of the disordered graphene sheets due to the reduction at high temperature.

The stress–strain measurements in Figure 3d display the typical deformation behavior of an open honeycomb-like aerogel, including linear elastic behavior at low strain, a cell collapse-related stress reduction at intermediate strain, and a plastic yielding plateau with subsequent stiffening at high strain [29]. Indeed, previous work has shown that crosslinker made by PMVEMA and PEG can form ester bonds with hydroxyl moieties on CMC/GO, resulting in a stiffness increase of aerogel. It can be found that the pure CMC aerogel has a compression stress of 3.72 MPa, while with the incorporation of GO, the compression stress increased to 7.14 MPa. Furthermore, after the CMC/GO aerogel was crosslinked via PMVEMA/PEG, its mechanical strength continued increasing and reached 7.92 MPa (uncured) and 12.32 MPa (cured), respectively. These results indicated that the highly organized structures in the achieved aerogels enhanced mechanical performance properties.

To further confirm the strong interaction between CMC and GO, XPS measurements were carried out and the results are shown in Figure 5. The CMC spectrum can be fitted with three peaks at bending energies at 284.85, 286.32, 287.86 eV [37], which can be assigned to C–C/C=C, C–O, C–O–C bonds, respectively. Peaks for C–O and C–O–C in the CMC/GO aerogel (286.48 and 288.07 eV, respectively) and the L-CMC/GO aerogel (286.30 and 287.78 eV, respectively) experienced apparent variations, indicating a strong interaction between CMC, GO and the crosslinker [38]. After treatment, the peaks for C–O of L-CMC/rGO aerogel (reduced after curing at 105 °C for 12 h) suffered an obvious decrease, which may be caused by the GO reduction process at high temperature.

### 3.2. Adsorption Behavior toward MB

#### 3.2.1. Effect of GO Addition and Initial Concentration of MB

The effect of GO addition on the MB adsorption is shown in Figure 6a. The q_e_ value increased from 98.65 to 124.67 mg/g with the added amount of GO ranging from 1 to 10 *w*/*w*%, indicating that the GO contributed to the adsorption capacity of the L-CMC/rGO aerogels. This is because the hydrophilic hydroxyl groups and benzene ring structure of GO provided more adsorption sites for MB [35], resulting in strong interactions between aerogel and MB molecules.

The initial dye concentration is another major factor affecting the adsorption process. It was observed in Figure 6b that, as the initial concentration increased, the MB removal percentage remained unchanged at the beginning and then decreased. The adsorption efficiency reached 93.41% at 500 ppm. However, when further increasing the initial concentration, it quickly dropped below 90%. This is because a lesser number of adsorbent sites was available for the increased number of MB molecules.

#### 3.2.2. Effect of pH and Temperature

The effect of pH value on MB removal percentage was also studied, as shown in Figure 6c. The maximum adsorption efficiency was observed at neutral pH, due to the electrostatic interaction between cationic dye MB and the L-CMC/rGO aerogel with negative charged surface. In basic condition, the high concentration of OH^−^ would preferentially attract MB by the electrostatic attraction, generating hydroxylated MB molecules that decreased the adsorption capacity due to the electrostatic repulsion effect. When the pH is lower than 7, the adsorption capacity also decreased, mainly due to the positive potential on the surface of the aerogel, which reduced the electrostatic interaction between aerogel and MB.

The environmental temperature also affects the adsorption efficiency. As shown in Figure 6d, the influence of temperature was evaluated in the range from 25 to 45 °C and the maximum removal percentage of L-CMC/rGO aerogel was observed at RT (25 °C). This is probably because the thermal motion of the MB molecules accelerated at high temperature, leading to a faster adsorption–desorption equilibrium [21].

#### 3.2.3. Adsorption Kinetics

Kinetics prediction of the adsorption capacity and adsorption time plays an important role in revealing the adsorption mechanism. The adsorption kinetic data of the L-CMC/rGO aerogel were analyzed using the well-known pseudo-first-order model and pseudo-second-order model. The former one can be expressed as Equation (3):(3)$$\ln\left(q_{e}-q_{t}\right)=\ln q_{e}-k_{1}t$$
where *q*_e_ and *q*_t_ (mg/g) are the amounts of MB adsorbed at equilibrium and various times t (min). *k*_1_ (1/min) is the equilibrium rate constant of the pseudo-first-order model. The kinetics parameters can be calculated from the intercept and slope of the linear relationship of ln(*q*_e_−*q*_t_) versus t in Figure 7a.

The experimental values *q*_e_ did not agree with the calculated ones as obtained from the linear plots, and the values of the correlation coefficient (R^2^) were relatively low (Table 1), which demonstrated that the adsorption process of crosslinked CMC/rGO aerogel did not comply with the pseudo-first-order model.

Another kinetic model, the pseudo-second-order model, contains different kinds of adsorptions including external film diffusion, adsorption and internal particle diffusion. It is expressed as Equation (4):(4)$$\frac{t}{q_{t}}=\frac{1}{k_{2}q_{e}^{2}}+\frac{1}{q_{e}}t$$
where *k*_2_ (g/mg·min) represents the equilibrium rate constant of the pseudo-second-order model. The linear plots of *t*/*q*_t_ against t in Figure 7b showed an excellent agreement between experimental and calculated values at different initial concentrations. The values of R^2^ were particularly high (Table 1), which shows that the pseudo-second-order model is more suitable for the adsorption of L-CMC/rGO aerogel and the process is controlled by the rate-determining step, chemisorption.

#### 3.2.4. Adsorption Isotherm

Adsorption isotherms are usually utilized to analyze the interactions between adsorbent and adsorbate. The Langmuir and Freundlich models shown in Figure 8 were examined for determining the adsorption mechanism of the L-CMC/rGO aerogel. The Langmuir model can be expressed as Equation (5), which is suitable for the monolayer adsorption occurring on a homogeneous surface without subsequent interaction among the adsorbed molecules, while the Freundlich model is based on multilayer adsorption on heterogeneous surfaces and can be described as Equation (6):(5)$$\frac{c_{e}}{q_{e}}=\frac{c_{e}}{q_{m}}+\frac{1}{q_{m}K_{L}}$$
(6)$$\ln q_{e}=\frac{1}{n}\ln c_{e}+\ln K_{F}$$
where *c*_e_ (mg/L) represents the equilibrium concentration of MB; *q*_m_ (mg/g) is the maximum adsorption capacity of L-CMC/rGO10 aerogel; *K*_L_ (L/mg) is the Langmuir constant; *n* and *K*_F_ are the Freundlich constants. The parameters obtained from both the Langmuir and Freundlich models are shown in Table 2. R^2^ of the former one (0.9997) is higher than that of the latter one (0.9306), which indicated that the Langmuir model could well describe the adsorption behavior of MB on L-CMC/rGO aerogel.

### 3.3. Adsorption Mechanism

Overall, the adsorption mechanism associated with MB removal on the L-CMC/rGO aerogel distinctly follows several stages, including (1) the migration of dye molecules from the solution to the outer surface of the adsorbent, (2) the diffusion of dye molecules through the solution/solid interface, interaction among the dye and the adsorption sites localized on the adsorbent surface (inner surface) and (3) the diffusion and interpenetration of the dye molecules into the solid pores. Synthesized L-CMC/rGO aerogel formed perfect 3D networked structure and dye molecules are readily embedded into this structure due to the hydrogen bonding interaction and ester bonding. The positive charge present on the N atom of MB interacts with the COO^−^ and OH groups on the aerogel network. GO as the active adsorption sites provided an advantageous condition for attracting more MB molecules and consequently improved the rate of adsorption capacities in theory. Moreover, the ion exchange process, hydrogen bonding, π–π interaction and electrostatic interaction [19,39,40] of L-CMC/rGO aerogel and MB molecules are also responsible for the MB removal.

### 3.4. Recyclability Experiments

Recyclability of the L-CMC/rGO aerogel was investigated in terms of removal percentage by adsorption-desorption cycles and the results are shown in Figure 9. It was observed that the aerogel retained 97.43% of its initial adsorption efficiency even after five cycles of reuse, which is because of the template-directed growth method with the crosslinking system, resulting in excellent structure stability and recyclability. The adsorption capacity slightly decreased and it took more time to remove the same percentage of MB in the subsequent recycling process (cycle more than 5). The probable reasons are the partial degradation of the aerogel during the recovery and desorption process, and the desorption of MB from the L-CMC/rGO aerogel was not complete, resulting in the decrease of available adsorption sites in the subsequent process. The other reason is possibly related to the reduction of the exfoliation degree of GO during the recycling process, which leads to the decrease of the specific surface area. In sum, the L-CMC/rGO aerogel with attractive properties (e.g., low cost, easy handling, excellent adsorption capacity and recyclability) encourage its potential application in wastewater treatment.

## 4. Conclusions

A novel and facile method was proposed to prepare a 3D highly aligned porous CMC/rGO aerogel with good structure stability and excellent adsorption properties for methylene blue (MB) removal via a template-directed freezing technique. FT-IR spectra showed the ester bond between CMC/GO and PMVEMA/PEG. The XRD, Raman and XPS spectra also supported the interaction between the base components of the synthesized aerogel. In addition, it was verified that the increase of GO addition strengthened the mechanical properties of the crosslinked aerogels. Moreover, the interaction between CMC and GO was enhanced due to the introduction of PMVEMA and PEG as crosslinking agents. The composite aerogel with 10 *w*/*w*% GO showed the highest MB adsorption capacity of 520.67 mg/g. The kinetics and isotherm studies of the resulting adsorbent showed that the adsorption obeyed the pseudo-second-order model and the Langmuir adsorption model, respectively, and adsorption is a spontaneous process. As a result, recycling and reuse experiments indicated the synthesized composite aerogel exhibited the ability to be used for consecutive adsorption/desorption cycles. The straightforward synthesis, nontoxicity, high adsorption capacity and recyclability make the crosslinked CMC/rGO aerogel a promising adsorption material for removing organic dyes from liquid and even larger scale.

## Figures and Tables

**Figure 1 polymers-12-02219-f001:**
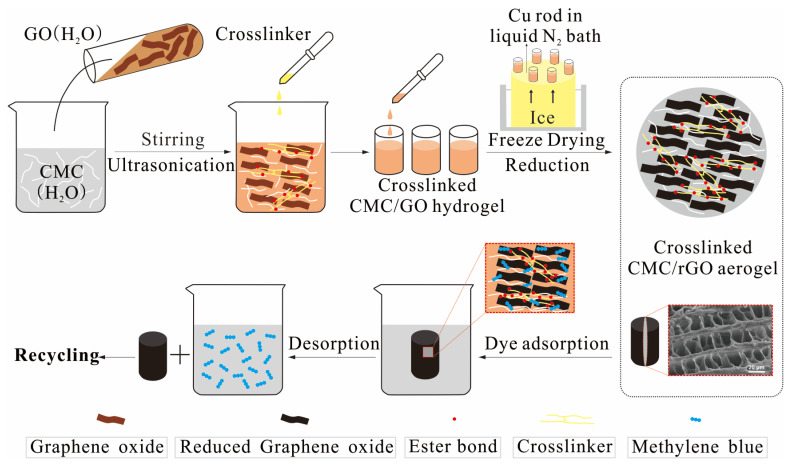
Schematic of the fabrication protocols of the crosslinked carboxymethyl cellulose (CMC)/reduced graphene oxide (rGO) aerogel.

**Figure 2 polymers-12-02219-f002:**
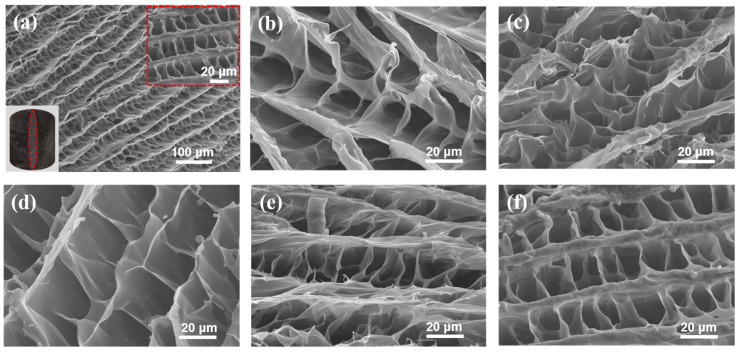
(**a**) shows the macro morphology and micro morphologies of the L-CMC/rGO aerogel with 10% GO addition, (**b**)–(**f**) show the structural morphologies of the L-CMC/rGO aerogels with 1%, 3%, 5%, 7% and 10% GO addition.

**Figure 3 polymers-12-02219-f003:**
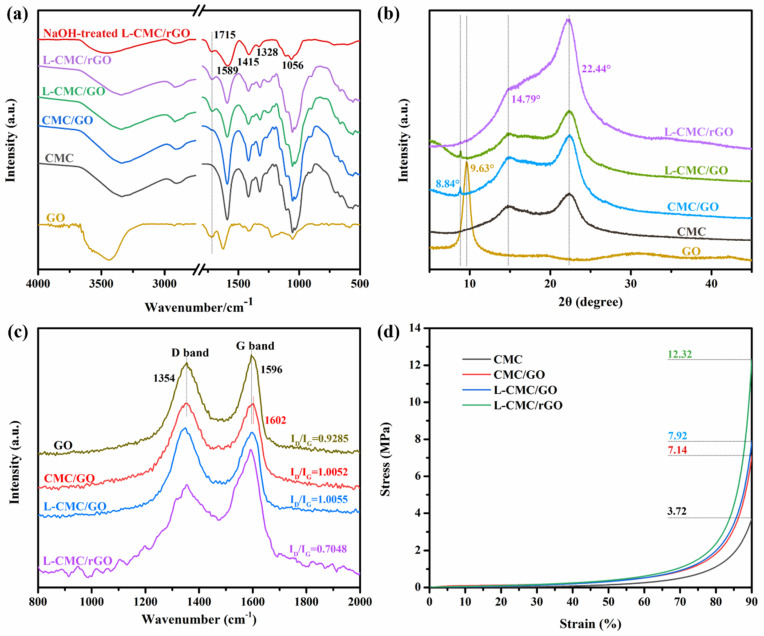
(**a**) FT-IR spectra; (**b**) XRD pattern; (**c**) Raman spectra and (**d**) stress–strain diagram form compression test of initial components and prepared aerogels.

**Figure 4 polymers-12-02219-f004:**
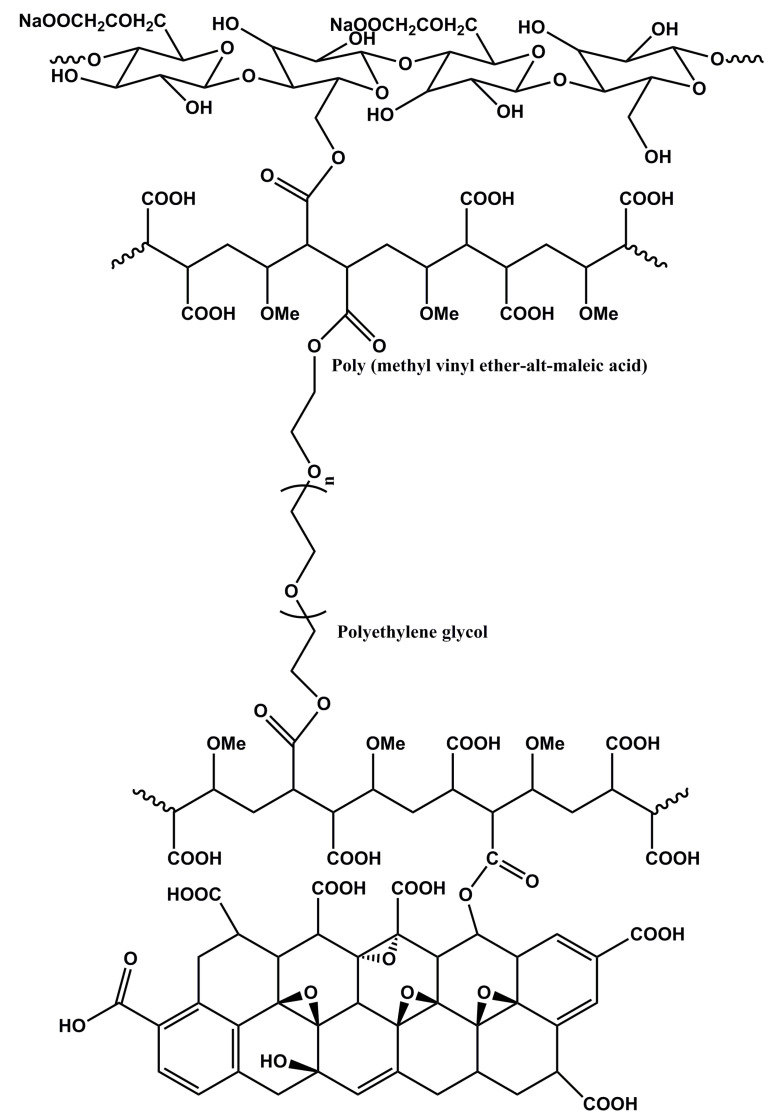
Molecular structure for poly (methyl vinyl ether-alt-maleic acid) (PMVEMA)/polyethylene glycol (PEG) crosslinking agent between CMC and GO.

**Figure 5 polymers-12-02219-f005:**
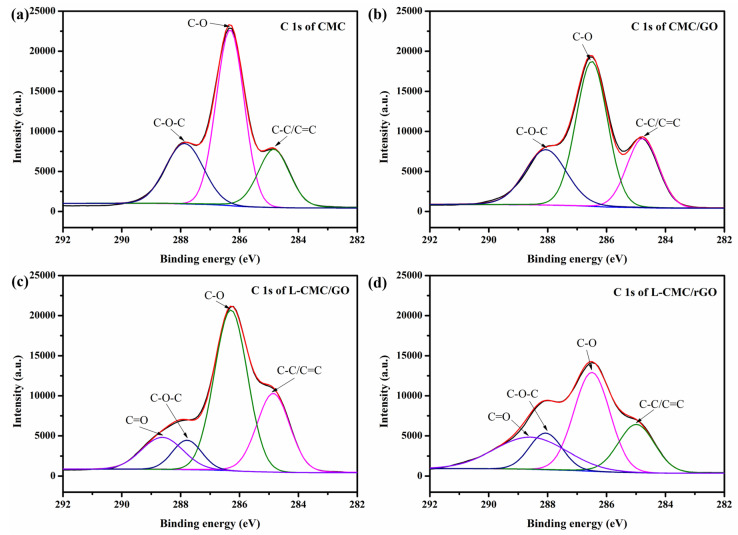
XPS C 1s spectra of (**a**) CMC, (**b**) CMC/GO, (**c**) L-CMC/GO and (**d**) L-CMC/rGO aerogels.

**Figure 6 polymers-12-02219-f006:**
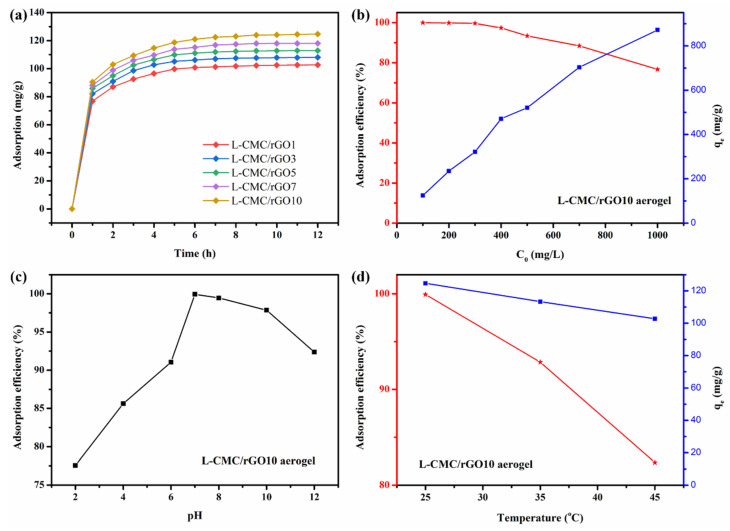
Effect of GO addition (**a**); initial concentration of methylene blue (MB) (**b**); pH (**c**) and temperature (**d**) for MB adsorption on L-CMC/rGO aerogel.

**Figure 7 polymers-12-02219-f007:**
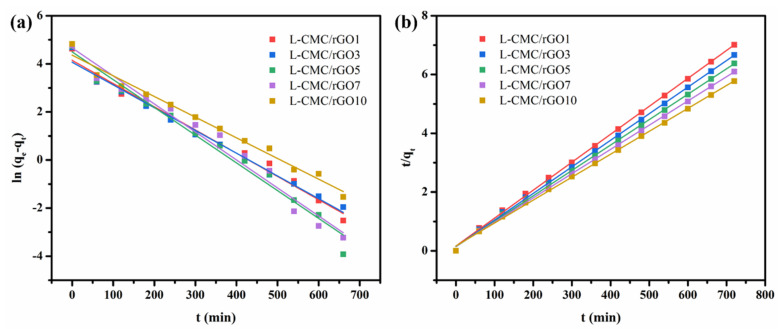
Pseudo-first-order model (**a**) and pseudo-second-order model (**b**) of MB adsorption on L-CMC/rGO aerogel.

**Figure 8 polymers-12-02219-f008:**
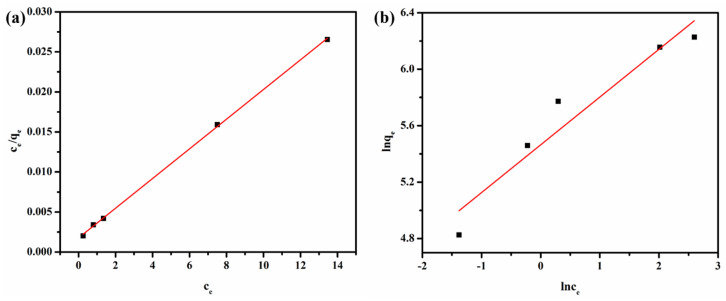
Langmuir (**a**) and Freundlich (**b**) curve of MB adsorption on L-CMC/rGO10 aerogel.

**Figure 9 polymers-12-02219-f009:**
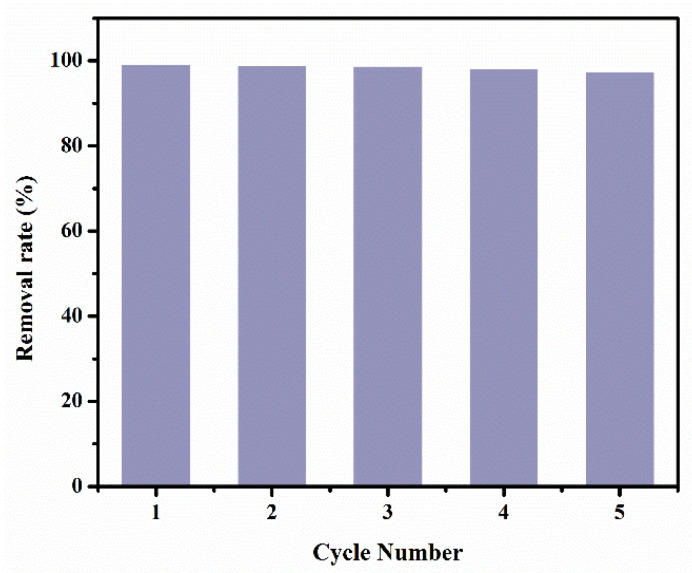
Recyclability plots of MB adsorption efficiency on L-CMC/rGO aerogel.

**Table 1 polymers-12-02219-t001:** Parameters of adsorption kinetics toward MB by L-CMC/rGO aerogel.

Samples	Pseudo-First-Order			Pseudo-Second-Order		
	*q*_e_ (mg/g)	*k*_1_ (g/mg·min)	R^2^	*q*_e_ (mg/g)	*k*_2_ (g/mg·min)	R^2^
L-CMC/rGO1	63.22	0.00966	0.9851	105.15	0.00056	0.9993
L-CMC/rGO3	58.54	0.00946	0.9875	110.62	0.00057	0.9994
L-CMC/rGO5	89.86	0.01154	0.9797	115.61	0.00053	0.9993
L-CMC/rGO7	106.10	0.01165	0.9754	121.40	0.00044	0.9991
L-CMC/rGO1	78.46	0.00860	0.9854	128.21	0.00037	0.9990

**Table 2 polymers-12-02219-t002:** Parameters of adsorption isotherms toward MB by L-CMC/rGO10 aerogel.

T/°C	Langmuir Adsorption Model			Freundlich Adsorption Model		
	*q*_m_ (mg/g)	*K*_L_ (L/mg)	R^2^	*K*_F_ (mg/g)(L/mg)^1/*n*^	*n*	R^2^
25	540.54	1.051	0.9997	236.014	2.955	0.9306

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
