# Peer review of "Directionally-Grown Carboxymethyl Cellulose/Reduced Graphene Oxide Aerogel with Excellent Structure Stability and Adsorption Capacity"

_polymers, 2020, doi:10.3390/polym12102219_

Round 1
Reviewer 1 Report
Revision for polymers-935491
Directionally-grown PMVEMA/PEG crosslinked carboxymethyl cellulose/reduced graphene oxide aerogel with excellent structure stability and adsorption capacity for methylene blue
In this work the authors prepare a carboxymethyl cellulose/reduced graphene oxide (CMC/rGO) aerogel, crosslinked by means poly (methyl vinyl ether co maleic acid)/poly (ethylene glycol) (PMVEMA/PEG). The growth of the crosslinking is directional thanks to via a directional freezing technique. The result is an aerogel with very goo structure stability and adsorption capacity.
The experimental methodology is well conducted, and all the conclusions are supported by the characterization methods used. The manuscript is well written and in general the document is suitable for publication in Polymers after only minor changes.
My suggestions are:
- The title is a bit too long, please suggest another one. Maybe the methylene blue is not necessary to appear in the title (?), please check/change
- MB is used as acronym for methylene blue. It is only described in the abstract. It is necessary to write it again in the introduction (and also maybe in conclusions). The same for RT room temperature, it is not described. Please check if this happens for any other acronym. In general, it is not enough if the acronym is described in the abstract…
- Figure 1: the authors use the same symbol (color) for GO and for rGO; it should be distinguished.
- The reduction process for GO is not described. Does this happen during the treatment at 105oC? For how long did this process take place? Please, check and add some reference.
- In general, the authors do not justify properly the discussion of results. In the introduction part there appear 3 main references for their work (ref 29-31) and they do not appear during the discussion. Please, include references in the results section, that provide a validation of your results.
With these modifications, the manuscript is ready for publication.
Author Response
Responses to Reviewer Comments for Manuscript: "Directionally-grown PMVEMA/PEG crosslinked carboxymethyl cellulose/reduced graphene oxide aerogel with excellent structure stability and adsorption capacity for methylene blue" (polymers-935491)
We appreciate the time and efforts by the editor and referees in reviewing this manuscript! The revised manuscript fully addresses the concerns raised by the review and strengthens the overall manuscript.
The changes have been made with “red color text” in the revised manuscript.
All our responses to the reviewers’ comments below are in red.
Reviewer Comments:
Reviewer #1: In this work the authors prepare a carboxymethyl cellulose/reduced graphene oxide (CMC/rGO) aerogel, crosslinked by means poly (methyl vinyl ether co maleic acid)/poly (ethylene glycol) (PMVEMA/PEG). The growth of the crosslinking is directional thanks to via a directional freezing technique. The result is an aerogel with very goo structure stability and adsorption capacity.
The experimental methodology is well conducted, and all the conclusions are supported by the characterization methods used. The manuscript is well written and in general the document is suitable for publication in Polymers after only minor changes.
My suggestions are:
1. The title is a bit too long, please suggest another one. Maybe the methylene blue is not necessary to appear in the title (?), please check/change
Response:
Thanks for the suggestion, we have modified this title, according to the reviewer suggestion, to be “Directionally-grown carboxymethyl cellulose/reduced graphene oxide aerogel with excellent structure stability and adsorption capacity” (lines 2-4).
- MB is used as acronym for methylene blue. It is only described in the abstract. It is necessary to write it again in the introduction (and also maybe in conclusions). The same for RT room temperature, it is not described. Please check if this happens for any other acronym. In general, it is not enough if the acronym is described in the abstract…
Response:
Thanks for the suggestion. We have given the full name of methylene blue and room temperature when they first appeared in every section (see lines 58-60, 107-108, and 349-351). In addition, we carefully checked the manucript to ensure that the same mistakes have been corrected. The revised sentences are as follows:
“For example, Chen et al.[19] synthesized a reusable 3D agar/GO composite aerogel by a vacuum freeze-drying method and evaluated its potential as an adsorbent of methylene blue (MB).” (lines 58-60)
“First, CMC powder was dissolved in DI water (2 wt%) at room temperature (RT)with agitation for 12 h.” (lines 108-109)
“A novel and facile method was proposed to prepare a 3D highly aligned porous CMC/rGO aerogel with good structure stability and excellent adsorption properties for the methylene blue (MB) removal via a template-directed freezing technique.” (lines 350-352)
- Figure 1: the authors use the same symbol (color) for GO and for rGO; it should be distinguished.
Response:
Thanks for the comment. We modified Figure 1 to make it more distinguishable, and the GO and rGO are now marked with brown and black colors, respectively. (see the figure below and Figure 1 in the revised manuscript)
Figure 1. Schematic of the fabrication protocols of the crosslinked CMC/rGO aerogel.
- The reduction process for GO is not described. Does this happen during the treatment at 105oC? For how long did this process take place? Please, check and add some reference.
Response:
Thanks for the comment. The GO reduction process occurred during the treatment at 105oC for 12 h. We have added this information into section 2.2. Moreover, the related references have also been given. (lines 119-120)
The revised sentence is as follows: “Finally, samples were cured at 105oC for 12 h to obtain crosslinked CMC/rGO (L-CMC/rGO) aerogels[32].”
[32] Xiang, C.; Wang, C.; Guo, R.; Lan, J.; Lin, S.; Jiang, S.; Lai, X.; Zhang, Y.; Xiao, H. Synthesis of carboxymethyl cellulose-reduced graphene oxide aerogel for efficient removal of organic liquids and dyes. Journal of Materials Science 2018, 54, 1872-1883, doi:10.1007/s10853-018-2900-5.
- In general, the authors do not justify properly the discussion of results. In the introduction part there appear 3 main references for their work (ref 29-31) and they do not appear during the discussion. Please, include references in the results section, that provide a validation of your results.
Response:
Thanks for the comment. The references 29, 30, and 31(in the original manuscript) have been included in the results section (lines 224-227, lines 190-192, and lines 197-201, respectively). The revised contents are as follows:
“The stress-strain measurements in Figure 4d display the typical deformation behavior of an open honeycomb-like aerogel, including linear elastic behavior at low strain, a cell collapse-related stress reduction at intermediate strain, and a plastic yielding plateau with subsequent stiffening at high strain[29].”
“The L-CMC/rGO aerogel crosslinked by PMVEMA/PEG system[30] (Figure 3) was previously reported to occur through an esterification reaction[34].”
“In the NaOH treated sample, the peak at 1589 cm-1 was attributed to the carboxylate formed because of the neutralization of the unreacted carboxylic acid groups and the sodium hydroxide, and the remaining signal at 1715 cm-1 was solely confirmed the establishment of ester cross-linkages between cross-linker and CMC/GO, which were previously reported by Goetz et al[27,31].”
[29] Wicklein, B.; Kocjan, A.; Salazar-Alvarez, G.; Carosio, F.; Camino, G.; Antonietti, M.; Bergstrom, L. Thermally insulating and fire-retardant lightweight anisotropic foams based on nanocellulose and graphene oxide. Nat Nanotechnol 2015, 10, 277-283, doi:10.1038/nnano.2014.248.
[30] Liang, L.; Huang, C.; Hao, N.; Ragauskas, A.J. Cross-linked poly(methyl vinyl ether-co-maleic acid)/poly(ethylene glycol)/nanocellulosics foams via directional freezing. Carbohydr Polym 2019, 213, 346-351, doi:10.1016/j.carbpol.2019.02.073.
[31] Liang, L.; Zhang, S.; Goenaga, G.A.; Meng, X.; Zawodzinski, T.A.; Ragauskas, A.J. Chemically Cross-Linked Cellulose Nanocrystal Aerogels for Effective Removal of Cation Dye. Front Chem 2020, 8, 570, doi:10.3389/fchem.2020.00570.

Reviewer 2 Report
The work is interesting and can be published based on the major revision required data, which are asked below.
- The author should provide the scheme of chemical interaction between PMVEMA/PEG and CMC/GO. It means the bond formation mechanism due to the interaction of functional groups. It is also necessary need to show the relation of dye separation with new bond formation.
- Authors mentioned that the characteristic C=C cannot be clearly observed, which revealed the strong interaction between CMC and GO. It is better to explain with proper references that why strong strong interaction disappear the C=C peak? Generally strong interaction creates new peaks or strong peak.
Author Response
Responses to Reviewer Comments for Manuscript: "Directionally-grown PMVEMA/PEG crosslinked carboxymethyl cellulose/reduced graphene oxide aerogel with excellent structure stability and adsorption capacity for methylene blue" (polymers-935491)
We appreciate the time and efforts by the editor and referees in reviewing this manuscript! The revised manuscript fully addresses the concerns raised by the review and strengthens the overall manuscript.
The changes have been made with “red color text” in the revised manuscript.
All our responses to the reviewers’ comments below are in red.
Reviewer Comments:
Reviewer #2: The work is interesting and can be published based on the major revision required data, which are asked below.
Comments:
1. The author should provide the scheme of chemical interaction between PMVEMA/PEG and CMC/GO. It means the bond formation mechanism due to the interaction of functional groups. It is also necessary need to show the relation of dye separation with new bond formation.
Response:
Thanks for the comment. The crosslinking agent is synthesized by the reaction between PMVEMA and PEG, thus generating a crosslinker that is abundant with carboxyl groups. During the formation of the crosslinked aerogel, the crosslinking agent connected CMC and GO through ester bonds, thereby forming a networked structure. The formation of new bonds in the cross-linking process could give the aerogel better mechanical properties so as to ensure that it has excellent structure stability when applied to the separation of organic dye from wastewater. According to the reviewer’s suggestion, we have given the scheme of the chemical interaction between the crosslinker and CMC/GO, as shown below (Figure 3 in the revised manuscript):
Figure 3. Molecular structure for PMVEMA/PEG crosslinking agent between CMC and GO.
In addition, the adsorption of organic dyes mainly occurred by the formation of hydrogen bonding, π-π interaction, and electrostatic interaction between the porous L-CMC/rGO aerogel and dye molecules as reported by many publications [19,39,40], rather than the formation of new chemical bonds. A more detailed adsorption mechanism is explained in section 3.3:
“3.3. Adsorption mechanism
Overall, the adsorption mechanism associated with the MB removal on the L-CMC/rGO aerogel follows distinct several stages, including 1) the migration of dye molecules from the solution to the outer surface of the adsorbent, 2)the diffusion of dye molecules through the solution/solid interface, interaction among the dye and the adsorption sites localized on the adsorbent surface (inner surface) and 3) the diffusion and interpenetration of the dye molecules into the solid pores. Synthesized L-CMC/rGO aerogel formed perfect 3D networked structure and dye molecules are readily to embed into this structure due to the hydrogen bonding interaction and ester bonding. The positive charge present on the N atom of MB interacts with the COO− and OH groups on the aerogel network. GO as the active adsorption sites provided an advantageous condition for attracting more MB molecules and consquently improved the rate of adsorption capacities in theory. Moreover, ion exchange process, hydrogen bonding, π-π interaction, and electrostatic interaction [19, 39, 40] of L-CMC/rGO aerogel and MB molecules are also responsible for the MB removal.
[19] Chen, L.; Li, Y.; Du, Q.; Wang, Z.; Xia, Y.; Yedinak, E.; Lou, J.; Ci, L. High performance agar/graphene oxide composite aerogel for methylene blue removal. Carbohydr Polym 2017, 155, 345-353, doi:10.1016/j.carbpol.2016.08.047.
[39] Huang, T.; Shao, Y.-w.; Zhang, Q.; Deng, Y.-f.; Liang, Z.-x.; Guo, F.-z.; Li, P.-c.; Wang, Y. Chitosan-Cross-Linked Graphene Oxide/Carboxymethyl Cellulose Aerogel Globules with High Structure Stability in Liquid and Extremely High Adsorption Ability. ACS Sustainable Chemistry & Engineering 2019, 7, 8775-8788, doi:10.1021/acssuschemeng.9b00691.”
[40] Fu, J.; Chen, Z.; Wang, M.; Liu, S.; Zhang, J.; Zhang, J.; Han, R.; Xu, Q. Adsorption of methylene blue by a high-efficiency adsorbent (polydopamine microspheres): Kinetics, isotherm, thermodynamics and mechanism analysis. Chemical Engineering Journal 2015, 259, 53-61, doi:10.1016/j.cej.2014.07.101
- Authors mentioned that the characteristic C=C cannot be clearly observed, which revealed the strong interaction between CMC and GO. It is better to explain with proper references that why strong strong interaction disappear the C=C peak? Generally strong interaction creates new peaks or strong peak.
Response:
Thanks for the comment. We made a mistake here, and we have carefully made serious changes to this part, and the revised sentences (lines 187-192) are as follows:
While for the CMC/GO aerogel, the characteristic band of C=O of GO can be observed but not obvious, which revealed the successful introduction of GO. And the peak was not obvious may be due to the low amount of GO. The L-CMC/rGO aerogel (reduced after curing at 105oC for 12 h) crosslinked by PMVEMA/PEG system[30] (Figure 3) was previously reported to occur through an esterification reaction[34].
[30] Liang, L.; Huang, C.; Hao, N.; Ragauskas, A.J. Cross-linked poly(methyl vinyl ether-co-maleic acid)/poly(ethylene glycol)/nanocellulosics foams via directional freezing. Carbohydr Polym 2019, 213, 346-351, doi:10.1016/j.carbpol.2019.02.073.
[34] Huang, C.; Hao, N.; Bhagia, S.; Li, M.; Meng, X.; Pu, Y.; Yong, Q.; Ragauskas, A.J. Porous artificial bone scaffold synthesized from a facile in situ hydroxyapatite coating and crosslinking reaction of crystalline nanocellulose. Materialia 2018, 4, 237-246, doi:10.1016/j.mtla.2018.09.008.

Round 2
Reviewer 2 Report
The author addressed all of the required question in the author reply report. So, the present form of the manuscript can be published.